# Transcriptomic Signatures in Lung Allografts and Their Therapeutic Implications

**DOI:** 10.3390/biomedicines12081793

**Published:** 2024-08-07

**Authors:** Michael Tyler Guinn, Ramiro Fernandez, Sean Lau, Gabriel Loor

**Affiliations:** 1Division of Cardiothoracic Transplantation and Circulatory Support, Michael E. DeBakey Department of Surgery, Baylor College of Medicine, Houston, TX 77030, USA; michael.guinn@bcm.edu (M.T.G.);; 2Department of Bioengineering, Rice University, Houston, TX 77005, USA; 3Department of Biology, The University of Texas at Austin, Austin, TX 78712, USA

**Keywords:** ex vivo lung perfusion, transcriptomics, lung transplantation, therapeutics

## Abstract

Ex vivo lung perfusion (EVLP) is a well-established method of lung preservation in clinical transplantation. Transcriptomic analyses of cells and tissues uncover gene expression patterns which reveal granular molecular pathways and cellular programs under various conditions. Coupling EVLP and transcriptomics may provide insights into lung allograft physiology at a molecular level with the potential to develop targeted therapies to enhance or repair the donor lung. This review examines the current landscape of transcriptional analysis of lung allografts in the context of state-of-the-art therapeutics that have been developed to optimize lung allograft function.

## 1. Introduction

Ex vivo lung perfusion (EVLP) technology allows normothermic perfusion and ventilation of lung allografts, enabling extended lung assessment before transplantation, reduced ischemia time, and prolonged total lung preservation time [1,2]. This has allowed the expansion of the donor lung pool and reduced the geographic barriers to donor lung distribution while maintaining clinical outcomes similar to standard lung preservation techniques [3,4,5,6].

In addition, EVLP offers a platform to elucidate the mechanisms of transplant-related lung injury, develop biomarkers, and apply targeted therapies (Figure 1). EVLP provides an opportunity to apply advanced therapeutics to not only repair injured lungs, but also enhance organ performance beyond what is possible using standard organ preservation techniques. To do so, a key step is understanding gene expression patterns of the lung allograft prior to transplantation. However, the study of transcriptomic profiles of lung allografts on EVLP remains a nascent field.

There are multiple physiologic perturbations that occur during EVLP such as electrolytes disturbances [7], inflammatory cytokine production [8,9], increased cell death [10], and endothelial barrier dysfunction [11]. On the other hand, EVLP has demonstrated the ability to salvage marginal lungs for transplant and reduce the incidence of primary graft dysfunction [12]. A better understanding of the underlying molecular processes during EVLP could enhance our ability to target pathways to reduce injury and enhance graft function.

Transcriptomic analyses have allowed investigators to explore cell-specific gene expression profiles in various human tissues [13,14]. More recent RNA sequencing methods such as single-cell RNA sequencing (scRNAseq) can resolve gene expression profiles spatially and temporally. Additionally, scRNAseq allows the creation of organ atlases, which may provide the granular data necessary to establish clinical prediction models of organ function and subsequently guide clinical decisions [15,16]. Transcriptomics can also inform changes in gene expression in response to therapeutics on EVLP including pathway states and cellular dynamics [17].

The convergence of EVLP and transcriptomic technology has the potential to revolutionize our understand of lung allograft physiology and allow the development of targeted interventions on EVLP to enhance the quality of lungs and thereby improve lung transplant outcomes [18]. To understand future directions these paired technologies can generate, it is important to understand what is currently known regarding lung allograft physiology on EVLP and potential therapeutics. In this review, we explore the current landscape of molecular studies of lung allografts on EVLP and putative therapies to enhance lung function.

### 1.1. Transcriptomic Profiles of Lung Allografts

Ex vivo lung perfusion technology has been used in clinical lung transplantation for over a decade. It has successfully expanded the lung donor pool by salvaging marginal donor lungs that otherwise would have been discarded. Through normothermic perfusion and ventilation, marginal lungs undergo prolonged assessment and may even undergo reconditioning. There is evidence that EVLP, without any specific therapeutic addition, may improve lung function [19]. Despite the overwhelming clinical success of EVLP, the underlying molecular mechanisms by which marginal lungs are salvaged and reconditioned remain unknown. Transcriptomic analysis has the potential to greatly enhance our understanding of the underlying mechanisms of EVLP in organ assessment, preservation, and rehabilitation. Furthermore, greater insight into the molecular signatures of cold ischemia and ischemia-reperfusion injury (IRI) may lead to novel targeted therapies to reduce primary graft dysfunction (PGD) and improve outcomes. In recent years, there has been greater focus on this treatment paradigm. This focus comes at a crucial time, as EVLP technology is expanding and rapidly becoming available to all programs, large and small, in the US with lung bioengineering approaches and with the future prospect of donor lung management hubs, where donor lungs may undergo assessment, repair, and re-engineering [20]. With the increasing familiarity with EVLP and the extending preservation times, the next phase of organ enhancement is near. Although the transcriptomic analysis of donor lungs remains a nascent field, there are some insights that have been generated recently (Table 1) which we have summarized in Figure 2.

### 1.2. Inflammation Signatures in Lung Allografts

Several animal EVLP models have demonstrated a common underlying transcriptomic pattern, interestingly with upregulation of both pro- and anti-inflammatory pathways. In a rat lung EVLP model, Lonati and colleagues sought to understand the global biomolecular profile of EVLP, comparing perfusate molecular composition and transcriptomic analyses before and after acellular EVLP. Lung function was stable over 3 h of EVLP time without significant edema or gross pathologic change. Transcriptomic analysis revealed several enriched pathways including inflammation, stress response, and cell survival pathways after EVLP. Key upregulated pro-inflammatory genes included monocyte chemoattractant CCL2 and neutrophil chemokines CXCL1 and CXCL2, as well as key pro-inflammatory cytokine IL1b. There was also upregulation of genes involved in apoptosis such as Casp3 and BCL2. Interestingly, anti-inflammatory pathways were also upregulated during EVLP with increased expression of IL-10. Additionally, IL-4 expression was downregulated, suggesting dampening of the adaptive immune response. These results suggested a mixed effect of EVLP which upregulates some immune pathways while dampening others. The perfusate analysis correlated with the transcriptomic analysis, with elevated circulating levels of pro-inflammatory chemokines CCL2 and CXCL2 in the perfusate, which may suggest an overall balance towards a pro-inflammatory state during EVLP [21].

In recognition of the accumulation of pro-inflammatory mediators, metabolites, and other cellular byproducts during EVLP which may be detrimental to graft function, DeWolf and colleagues used a porcine EVLP model to study the physiologic and transcriptomic response to a 6-h EVLP run under various conditions. They compared four perfusate conditions: (1) no modification, (2) partial replacement of perfusate, (3) adult dialysis filter, and (4) pediatric dialysis filter. Notably, although dialysis did stabilize metabolic and electrolyte profiles, there was no difference in lung allograft performance during EVLP nor major gene expression profiles across the groups. In all groups, there was upregulation in inflammatory pathways with increased IL6, NF-kB, TLR, and iNOS signaling pathways, among others. Like the Lonati study, there was also upregulation of anti-apoptotic genes and cellular regeneration signaling, suggesting simultaneous activation of both inflammatory and healing pathways during EVLP [23].

In search of novel innate anti-inflammatory mechanisms that may be activated during EVLP, Elgharably and colleagues studied the expression of microRNA (miR) in human lungs on EVLP. MicroRNAs are small non-coding RNA molecules that regulate gene expression by either degrading or inhibiting translation of target mRNA. They have been shown to regulate immunological responses in various lung pathologies [24]. They found that miR17 and miR-548b were most highly expressed after EVLP and were generated primarily by alveolar epithelial cells. These two miRNAs have multiple mRNA targets and have been shown to possess anti-inflammatory properties and promote epithelial proliferation. Some of these mRNA targets included the key inflammatory cytokines CXCL2, IL1β, IL6, and TNFα. In summary, Elgharably and colleagues uncovered a novel mechanism showing EVLP modulates the inflammatory response of reperfusion and promotes epithelial healing. Modulating the innate expression of miRNA represents a potential therapy that can be applied during EVLP to engineer the organ for better performance [24].

In a series of studies, the Toronto group has investigated the transcriptomic profiles of human lung allografts across various conditions. They initially evaluated the changes in gene expression over time during a 12 h EVLP run. A 12 h timepoint was chosen based on the premise that, to use EVLP as a therapeutic platform to modify and enhance organs, prolonged perfusion times are necessary. They found that, during the initial hours of EVLP, there was a reduction in immune cell pro-inflammatory signaling pathways which they attributed to the flushing out of leukocytes. They also found an early increase in endothelial cell markers of inflammation and apoptosis pathways that decreased over time. By three hours of perfusion time, cell viability and survival pathways were upregulated, which correlated with their prior work demonstrating cellular tight junction recovery with EVLP. This work was notable for a transcriptomic signature characterized by initial cell death pathways giving way to survival and regeneration pathways over prolonged perfusion times. The findings also suggested prolonged EVLP times are safe and subsequent therapeutic applications feasible [27].

### 1.3. DBD vs. DCD Lung Transcriptomic Signatures

While EVLP has been instrumental in expanding the donor lung pool by salvaging marginal donor lungs, approximately 30% of lungs evaluated on EVLP are found to be unsuitable for transplant [29]. Many of the lungs evaluated on EVLP are from DCD donors where a greater understanding of the molecular mechanisms of DCD lungs on EVLP may uncover opportunities to enhance their reconditioning and improve the utilization rates. To that end, Stone and colleagues evaluated the ability of EVLP with and without the addition of an adenosine A2A receptor (A_2A_R) agonist to recondition DCD lungs in a mouse EVLP model. Compared to a control group undergoing cold static preservation, the EVLP group had downregulated inflammatory pathways such as IL-1, IL-8, and IL-17 signaling pathways. The group that underwent treatment with A_2A_R agonist had an even more significant reduction in pro-inflammatory pathways. Along with the transcriptomic level changes, the EVLP groups had improved graft function and reduced pulmonary edema compared to the control group. This study identified a mechanistic underpinning to the clinical practice where EVLP reconditions DCD lungs and improves their utilization in clinical lung transplantation. It also identified a potential targeted intervention that is applied during EVLP to enhance graft performance and improve recipient outcomes [22].

### 1.4. Transcriptomic Profiles of Human Lung Allografts

The Toronto group compared transcriptomic profiles in DBD vs. DCD lungs, with and without EVLP. The study used lungs that were used in clinical transplantation. Compared to the DCD group, the DBD group had upregulated inflammation pathways including IL6, HMGB1, and p38 MAP kinase signaling pathways. Overall, DBD lungs had upregulated inflammatory cytokine pathways while DCD lungs had upregulated cell death pathways. Notably, EVLP had a different transcriptomic response in DBD vs. DCD lungs. In DBD lungs, EVLP was associated with a pro-inflammatory signature including upregulated IL1β, IL6, CCL20, and PTX3 signaling pathways. Remarkably, the application of EVLP in DCD lungs did not result in significant differentially expressed genes. It may be that the DCD-specific transcriptomic signature overwhelmed the EVLP signal. However, a potential limitation of the study was the use of microarrays rather than next generation sequencing. A true unbiased transcriptomic analysis might have altered the results, particularly the effect of EVLP on DCD lungs [26].

Finally, the Toronto group compared gene expression profiles in DBD lungs undergoing EVLP vs. those that went straight to transplant. The objective was to identify common signaling pathways that could be targeted during EVLP to recondition, repair, and improve graft function. Using microarrays, they found 27 signaling pathways commonly enriched in both groups. Like prior data, inflammation and apoptosis pathways were most significantly upregulated in both groups, including TNF and IL1β pathways along with TLR/MyD88 signaling, suggesting activation of the innate immune system via damage-associated molecular patterns (DAMPs). There was also downregulation of metabolism and protein synthesis pathways in both groups. Notably, in the EVLP group, there was upregulation of vascular processes such as adherens junction organization and negative regulation of cell death in epithelial and endothelial cells, suggesting increased epithelial integrity. This work identified several common transplant-related lung injury pathways that can be blocked on EVLP [25].

### 1.5. Insights from Single-Cell RNA Sequencing in Lung Allografts

What was lacking in the above studies was granular resolution of gene expression patterns at specific single-cell resolution. Gouin and colleagues used single-cell RNA sequencing to obtain cell-type specific transcriptomic profiles of human lungs treated with EVLP [28]. They analyzed 4 h and 10 h timepoints to assess the response to a typical EVLP run as well as a prolonged timepoint, setting the stage for EVLP as a therapeutic platform and for logistical improvements such as time shifting. They focused their analysis on the alveolar epithelial, endothelial, myeloid, and lymphoid cell compartments as they are the most clinically relevant and can potentially be targeted with specific therapies. The alveolar epithelial cell response was characterized by increased inflammatory signals such as upregulated neutrophil and monocyte chemokine expression, and downregulated cytoskeleton activity which may implicate barrier dysfunction. At the 4 h mark, there was upregulated cell viability signaling, which is concordant with the Toronto data; however, by 10 h, there was evidence of mitochondrial dysfunction with decreased metabolism pathways including oxidative phosphorylation and amino acid metabolism. The endothelial cell response was uniform across the 10 h perfusion time with upregulated expression of numerous innate immune cell chemokines as well as IL1β response elements. There was also upregulated endothelin pathway expression, which may also implicate endothelial barrier dysfunction during EVLP. The myeloid cell compartment demonstrated differential responses depending on cell type, with resident alveolar macrophages (AM), monocytes, and other phagocytic cells having distinct signatures. The AMs showed upregulated inflammatory pathways with expression of monocyte chemokines at the 4 h mark only, while monocyte and monocyte-derived macrophages demonstrated increased inflammatory signals throughout the 10 h perfusion time. Interestingly, the lymphoid cells showed blunted activation signals and even suppression of immune effector function. However, the role of donor-derived adaptive immune cells in the host response to lung transplantation has not been well defined and remains unclear. The authors then assessed the relative contribution of pro-inflammatory mediators by each compartment, finding a distribution of roles across cell types. Epithelial cells predominantly expressed neutrophil chemokine CXCL1 while endothelial cells primarily expressed classical monocyte chemokine CCL2 as well as IL6 [30,31]. The monocyte-derived macrophages, which reside in the interstitium and remain in the lung despite clinical vascular flushing, expressed key inflammatory mediators in ischemia reperfusion injury such as IL1β and TNFα [28]. Based on this study, complementary therapies on EVLP can be used for specific compartments, such as an intratracheal treatment to target alveolar epithelium and alveolar macrophages, and an intravascular treatment to target the endothelial compartment.

Given recent advances in EVLP and molecular tools, lung transplantation is poised to utilize high-throughput transcriptomic methods to elucidate novel molecular mechanisms of transplant-related lung injury and create a landscape of organ profiles on a spectrum of healthy-to-non-salvageable organs. This will ultimately allow targeted interventions to enhance and even re-engineer donor lungs. Achieving this aim would be important to convert diseased lungs to healthy lungs and lay a path not only to increase the organ pool but also to enhance organ function and improve lung recipient outcomes [32,33,34].

## 2. Leveraging EVLP as a Platform to Repair, Recondition, and Re-Engineer Lung Allografts

The study of transcriptional changes in lung allografts during EVLP and in response to ischemia reperfusion injury has opened the door to the development of targeted therapies to repair marginal organs, mitigate IRI, and enhance graft function (Table 2). While this area remains a nascent field, and we have yet to develop the tools for on-demand targeted therapeutics in clinical EVLP use, there are numerous pre-clinical studies spanning the range of strategies, including physical circuit-level modifications, pharmacologic treatments, cellular and molecular therapies, and finally genetic engineering (Figure 3). In addition to its role in identifying putative novel therapies, transcriptomic analysis of the response to the therapies may uncover the mechanisms of action on a gene expression level, allowing a previously unknown depth of understanding regarding how each therapy affects the lung allograft.

### 2.1. Physical Therapies on EVLP Circuit

At the simplest level, the perfusion system itself can be considered a therapeutic strategy. By preserving the donor lungs in a physiologic state with normothermic perfusion and ventilation, ischemia time, which is the nemesis of organ transplantation, is thereby reduced. In the INSPIRE trial, which randomized donor lung management to portable EVLP using the Transmedics OCS Lung device vs. standard cold preservation, there was almost a 50% reduction in high-grade PGD in the EVLP group compared to the standard group [12].

Additional physical therapies include leukocyte filters that trap passenger leukocytes that remained in the donor lung despite intravascular flushing that occurs during the procurement process. The emerging literature has shown that circulating donor leukocytes damaged lung allografts through activation of caspase-1-induced pyroptotic cell death during EVLP. However, when leukocyte filters were added during EVLP, lung function improved during EVLP and after transplantation [39]. Additional work in a pig transplant model has shown that the filtering of donor circulating leukocytes on EVLP reduced the migration of donor-derived leukocytes into recipient lymph nodes, which in turn reduced direct allorecognition and the infiltration of recipient T cells into the allograft [40]. This simple solution may have downstream benefits in dampening the alloimmune response.

As described in the section above, lung allografts on EVLP upregulate inflammatory signaling pathways which result in the production of pro-inflammatory cytokines that are detectable in the perfusate. Our group and others have shown higher levels of circulating cytokines correlated with worse lung transplant outcomes including a higher incidence of high-grade PGD [65,66]. Cytokine adsorption has been studied as a simple technique to reduce the inflammatory milieu and improve graft function. Iskender and colleagues have shown in a porcine EVLP model that cytokine adsorption improved physiological parameters and pulmonary edema on EVLP compared to control [37]. Furthermore, when the lungs were transplanted, those in the treatment group demonstrated improved graft function including improved compliance compared to control [36]. Cytokine adsorption has also been used to rescue damaged donor lungs. Ghaidan and colleagues used a porcine transplant model where the donor pigs were treated with lipopolysaccharide (LPS) to induce ARDS prior to lung harvest. The donor lungs were then placed on EVLP where the treatment group underwent cytokine adsorption prior to transplant. Compared to control, the cytokine adsorption group demonstrated improved graft function with less PGD [38].

Despite the clinical use of EVLP for over a decade, there remains a lack of consensus on the ideal composition of the perfusate. This is evident when comparing the two most widely used EVLP protocols: the Toronto and OCS protocols. The Toronto protocol uses an acellular perfusate while the OCS protocol uses a cellular solution supplemented with packed red blood cells. Microarray data have shown a variable gene expression signature between cellular and acellular perfusates, with reduced expression of pro-inflammatory pathways such as TNF with the use of cellular perfusates [35]. Additionally, as interest grows to increase perfusion times for therapeutic or logistical reasons, optimization of the perfusate to sustain the organ for prolonged periods is a major research target. The Pittsburgh group has studied the effect of oxygen delivery at various concentrations through the membrane oxygenator during EVLP on functional outcomes in a rat model of lung transplantation. They found that a perfusate oxygen concentration of 40% led to the best post-transplant functional outcome compared to lower or higher concentrations [41].

Additional therapies on EVLP circuits that have been tested include the use of hypothermic EVLP to reduce the production of inflammatory cytokines [44]. Transcriptomic methods could be used to probe organs under these physical therapies to elucidate what genes or pathways are altered, which may allow for a rational design of future therapies with a greater impact on graft function.

### 2.2. Pharmacologic Therapies

Inherent in EVLP systems is the ability to treat the organ via both intratracheal and intravascular routes to target specific anatomical compartments, such as the alveolar or endovascular spaces. As discussed previously, scRNAseq has uncovered cell-specific responses on EVLP, which can be considered a form of reperfusion. Accordingly, complementary treatments can be delivered, for instance to alveolar epithelial cells and endovascular cells.

In a porcine lung transplant model where gastric acid aspiration was induced in the donor pigs, lung lavage with exogenous surfactant replacement on EVLP reduced markers of inflammation and lung injury while restoring lung function after transplant [45]. Nebulized N-acetylcysteine, which serves as an antioxidant, has also been found to improve oxygenation and reduce ischemic injury in a porcine transplant model where donor lungs underwent an extended 24 h period of cold static preservation [46]. Hydrogen gas has also been shown to reduce production of pro-inflammatory cytokines and increase the generation of antioxidants that resulted in reduced apoptosis, lower histologic signs of injury, and less pulmonary edema in a porcine DCD lung transplant model [43].

Neutrophils are the end-effector cells for ischemia reperfusion injury with secretion of neutrophil elastase representing a major mechanism of injury. Accordingly, targeting neutrophil elastase through delivery of a neutrophil elastase inhibitor in the perfusate has been shown to reduce injury and improve lung function [47]. Additional perfusate additives that have been shown to reduce lung injury and improve graft function include antioxidants alpha-1 antitrypsin and pyrrolidine dithiocarbamate, adenosine receptor modulators, and complement inhibitors such trimetazidine, steroids, and aspirin [7,59,60,61,62,63,67].

### 2.3. Cellular and Molecular Therapies

Mesenchymal stem cells (MSCs) have been regarded as a near-panacea for organ dysfunction. Naturally, the use of MSCs has been explored in lung transplantation to reduce the impact of lung ischemia reperfusion injury and ameliorate PGD [52,68] The use of MSCs in EVLP, delivered both intratracheally and through the perfusate, has been shown to decrease inflammatory injury in human lungs, restore epithelial barrier function, and improve alveolar fluid clearance to reduce pulmonary edema [53,54]. The Toronto group has demonstrated improved retention of MSCs when delivered intravascularly rather than intratracheally, which resulted in a reduced inflammatory cytokine profile in the lung tissue [56,57]. They have also used MSC therapy during EVLP in a porcine lung transplant model and found ischemia reperfusion injury was attenuated compared to control [62].

One of the main clinical barriers to MSC therapy in solid organ transplantation is uncertain long-term risk related to multi-potent progenitor cell therapy in an immunosuppressed host [69]. It has been postulated that the immunomodulating properties generated by MSCs result from the secretions of factors acting in a paracrine manner. Therefore, a simpler and potentially safer solution to MSC therapy may be treatment with extracellular vesicles derived from MSCs. Exosomes exert their effects by reducing apoptosis, diminishing expression of pro-inflammatory cytokines, and promoting cellular regeneration, thereby improving end-organ function [70,71]. Lonati and colleagues used a rodent EVLP model and showed that MSC-derived exosome therapy preserved ATP levels, reduced barrier dysfunction, and upregulated gene pathways promoting resolution of inflammation and oxidative stress [64]. Other studies found a reduction in inflammatory markers such as IL-17, TNF-α, HMGB1, and CXCL1, and inhibition of neutrophil extravasation [55,65]. These therapies have myriad beneficial effects, yet the mechanism of action for these benefits remains unknown. Application of transcriptomic analysis could shed light on the molecular mechanisms by which stem-cell-derived therapy improves graft function.

### 2.4. Genetic Engineering

Genetic engineering can be applied to serve three main functions during EVLP: (1) restore normal physiology within a damaged organ [72], (2) modulate the microenvironment such as inhibiting an inflammatory response (e.g., stimulate macrophages towards M2 polarization) [73], and (3) add a novel function to the tissue of interest. These tools could be applied to the organ itself, or to exogenous cells that are then delivered to the lungs on EVLP. Transcriptomics could subsequently validate the success of these therapies by investigating gene expressions and pathway changes in donor lungs. To date, the state-of-the-art in donor organ management in transplantation is to prevent organ degradation, that is, to preserve the organs in the state in which they were found. Genetic engineering has the potential to change the paradigm to one in which the goal is to re-engineer the organ to perform better than the state in which it was found.

Cellular engineering can allow the dynamic control of protein expression and the use of cellular sensors to detect diseased tissues and serve as actuators to implement change in a specific tissue of interest [48,49,50,51,74]. These tools are often brought into the target cell with the aid of viruses or other transfection agents [75,76,77]. Once inside the cell, these synthetic biology tools are often integrated into the genome providing permanent function which could be used to aid organ quality [78]. The simplest level of gene circuits, i.e., overexpressing transgenes, has already started to be explored. The Toronto group has studied adenoviral human interleukin-10 (AdhIL-10) gene delivery during EVLP in a porcine lung transplant model. There was successful transfection with detectable levels of IL-10 in the perfusate at 12 h and in the recipient plasma samples. There was no evidence of systemic toxicity in the recipient throughout the 7-day observation period. Lung function was improved in the treatment group with lower levels of local tissue and systemic inflammation compared to control. Interestingly, the alloimmune response to the donor lung was blunted in the treatment group [58]. This pre-clinical work has established gene therapy as a viable treatment option during EVLP to enhance organ function by modulating both the innate and adaptive immune responses.

Additional gene therapy tools can knock-out, tune-up, or tune-down gene expression. This has historically been transcriptional factors but in more recent years has utilized the powerful engineering platform of CRISPR/Cas system for targeting RNA or DNA [79,80,81,82,83]. These tools are flexible as they can cause both temporary (e.g., transcriptional alterations) and permanent changes (e.g., DNA editing). For example, when a biomarker profile reveals up-regulation of pro-inflammatory genes, researchers can customize genetic engineering tools to downregulate these targets through transcriptional inhibition which provides a mechanism to optimize organ health. The Toronto group has recently explored this approach in a proof-of-concept study. They established the feasibility of CRISPR-mediated transcriptional upregulation of endogenous IL-10 expression in vitro and in vivo in a rat model of lung transplantation. While using a standard immunosuppression regimen, the group successfully delivered Cas9-based activators to alveolar epithelium with an adenoviral vector. They then demonstrated a dose-dependent upregulation of IL-10 production in the lung in vivo without alteration of other inflammatory cytokines that was sustained for up to 14 days with continued immunosuppression. Finally, donor lungs treated in this manner were used in an allogeneic rat transplant model where IL-10 was expressed at significantly higher levels without significant signs of inflammation compared to control [84]. This study has opened the doors to advanced donor lung modulation techniques with the potential to greatly improve lung transplant outcomes.

## 3. Conclusions

An ideal technology that can probe donated organs will provide a holistic picture of the organ’s health and deliver actionable information for subsequent improvement and optimization of the organ. Transcriptomic methods allow for a high-resolution analysis of molecular biology relevant to transplantation [85]. Transcriptomics possesses a unique advantage over other diagnostic methods like cytokine assays by providing a high-resolution analysis of gene pathway changes and cellular dynamics. In this context, real-time molecular profiling could be combined with physiologic parameters for more accurate clinical decision making.

Currently, the clinical use of EVLP for the past decade is primarily for the extended assessment of marginal donor lungs. The time is ripe to enter the next phase of EVLP to improve donor lungs. While the coupling of transcriptomics and EVLP remains in the investigative phase and has not yet reached clinical practice, its potential impact to further elucidate the molecular pathways underlying lung allograft dysfunction, predict graft function, and become a platform for targeted interventions is substantial. Despite remarkable advances in immunosuppression techniques and post-transplant care, lung recipients continue to suffer from the worst outcomes of all solid organ transplants. To date, median survival is only six years [86]. Primary graft dysfunction remains the primary cause of poor early outcomes and one of the strongest risk factors for long-term graft failure [87]. Furthermore, the donor lung utilization rate remains low, currently around 20% [88]. Coupling transcriptomic techniques and therapeutic interventions with EVLP provides the opportunity to reveal dysregulated molecular pathways that can be optimized to finally ameliorate PGD and salvage more marginal donor lungs.

In the future, adding metabolomic analysis to transcriptomics may provide additional actionable data for organ health assessment and development of therapeutic targets. Metabolic data can provide insight into potential pathways to optimize organ recovery and functional improvement such as improving mitochondrial bioenergetics [89]. Metabolic signatures can also be used as biomarkers to stratify based on organ health and need for therapeutic intervention [64].

The future of EVLP will be such that the donor lungs will be placed on EVLP after procurement and a real-time cellular and molecular-level analysis of organ health through “omics” analysis will be part of the standard assessment. This molecular blueprint will be compared to known organ atlases to determine the best therapeutic interventions to improve organ quality in real time on the EVLP platform. These therapies could take the form of pharmacologic agents, cell-based strategies, and genetic engineering tools. This platform is not limited to lung transplantation, as this concept can be applied to heart, liver, and kidney transplantation as well.

In summary, the coupling of transcriptomics and EVLP represents a potential synergistic platform to expand our knowledge of transplant biology and implement targeted therapeutics to drastically improve lung transplant outcomes. To make this strategy feasible, future work will incorporate multi-omics techniques into rapid diagnostic tools for real-time point-of-care decision making to individualize organ repair and modification strategies and usher in a new era in donor lung management.

## Figures and Tables

**Figure 1 biomedicines-12-01793-f001:**
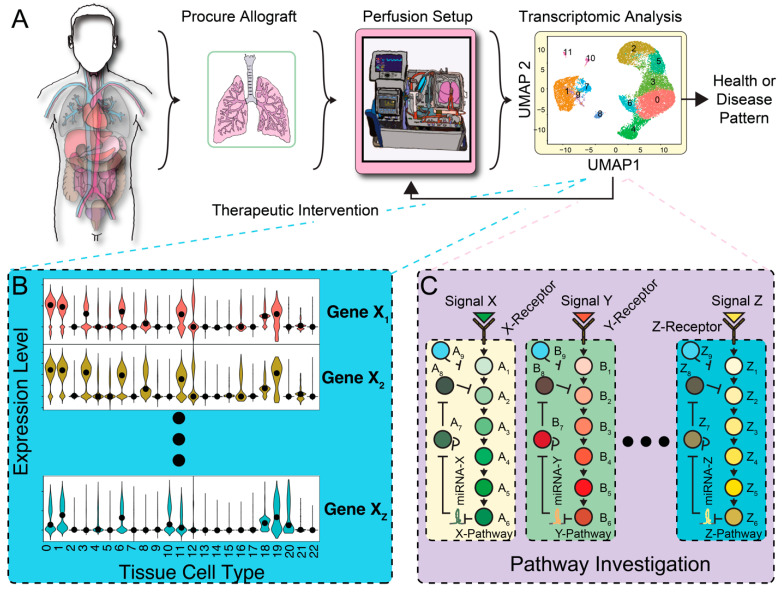
Coupling ex vivo perfusion systems with transcriptomic analysis. (**A**) Schematic illustration of proposed setup coupling ex vivo perfusion systems with transcriptomic technology like scRNAseq. The donor lungs are placed on EVLP for subsequent transcriptomic analysis. Numbers in scRNAseq graph represent different cell type clusters. (**B**) The downstream analysis produces gene expression profiles that can be compared to atlas databanks. (**C**) Subsequent pathway analysis informs individualized therapeutic strategies to repair and enhance organ function.

**Figure 2 biomedicines-12-01793-f002:**
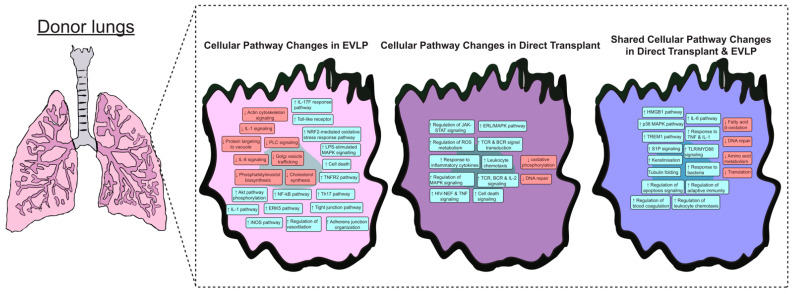
Pathway interpretation of genes and pathways in current literature. Findings were compiled across various literatures sources and stratified based on changes seen in lungs on EVLP, directly transplanted, or both. Up arrows indicate gene expression were upregulated while down arrows represent gene expression was down regulated.

**Figure 3 biomedicines-12-01793-f003:**
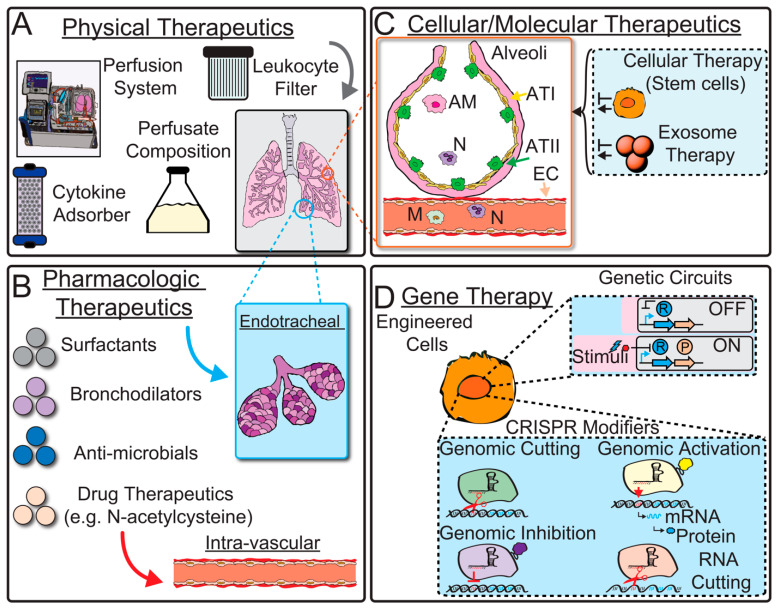
Therapeutic interventions for donor lungs on EVLP. (**A**) Schematic illustration of physical therapies. At the physical level, mechanical properties inherent to the EVLP circuit may be altered to improve donor lung function. (**B**) Schematic illustration of pharmacologic therapies. Pharmacologic treatments can be delivered intratracheally or intravascularly to target specific tissue compartments. (**C**) Schematic illustration of cellular and molecular therapies. Cellular and molecular therapies can be deployed to alter global inflammatory pathways. (**D**) Finally, gene therapy is the next phase in targeted therapeutics that can reengineer the donor lung for improved performance. Schematic illustration of repressor gene circuit OFF or ON with stimuli that can make a protein of interest. Additionally, CRISPR modifiers are shown with various molecular functions. AM, alveolar macrophage; N, neutrophil; M, monocyte; ATI, alveolar type I cell; ATII, alveolar type II cell; EC, endothelial cell.

**Table 1 biomedicines-12-01793-t001:** Transcriptomic studies on EVLP. Ex vivo lung perfusion (EVLP). Single-cell RNA sequencing (scRNAseq).

Study	Model	Analysis	Principle Finding	Tissue
Lonati et al. [21]	Rat	Microarray	Compared pre- vs. post-EVLP specimens and found both pro- and anti-inflammatory pathways activated simultaneously.	Lung
Stone et al. [22]	Murine	Microarray	In a donation-after-circulatory-death (DCD) lung model, the application of EVLP led to a decreased inflammatory profile compared to standard cold preservation.	Lung
De Wolf et al. [23]	Porcine	Bulk RNA sequencing	Porcine lungs on EVLP were subjected to various perfusate conditions—perfusate replacement, dialysis, and without modification. Compared to control, all EVLP groups displayed upregulated inflammatory, cell survival, and proliferation of connective tissue pathways, without a difference between EVLP groups.	Blood, Lung
Elgharably et al. [24]	Human	Bulk RNA sequencing	microRNA sequencing of rejected human lungs before and after EVLP showed upregulated miR-17 and miR-548b which are known to dampen downstream pro-inflammatory genes.	Perfusate solution, Lung
Wong et al. [25]	Human	Microarray	Donor lung gene expression profiles were compared in EVLP and clinical transplantation. Commonly altered pathways included upregulated inflammation and cell death pathways and downregulated metabolism pathways.	Lung
Baciu et al. [26]	Human	Microarray	Transcriptomic profiles were compared between DCD and donation-after-brain-death (DBD) lungs with and without EVLP. DBD lungs had upregulated inflammatory pathways while DCD lungs had upregulated cell death pathways.	Perfusate solution, Lung
Yeung et al. [27]	Human	Microarray	Transcriptomic profile investigation of rejected human lungs during 12 h of EVLP. Evidence of cellular recovery during EVLP was shown with cell death pathways upregulated early on while cell viability and survival pathways were upregulated at 3 h of perfusion time.	Lung
Gouin et al. [28]	Human	scRNAseq	Single-cell RNAseq of rejected human lungs at 4 h and 10 h of EVLP demonstrated unique patterns of gene expression across epithelial, endothelial, myeloid, and lymphoid compartments.	Perfusate solution, Lung

**Table 2 biomedicines-12-01793-t002:** Therapeutic interventions for lungs on EVLP. Ex vivo lung perfusion (EVLP). Perfluorocarbon-based oxygen carriers (PFCOC). Alpha-1 antitrypsin (A1AT). Bronchoalveolar lavage (BAL). Pulmonary vascular resistance (PVR). Mesenchymal stem cells (MSCs). Pulmonary artery pressure (PAP). Adenosine A2B receptor (A2BAR).

Study	Model	Finding
Dromparis et al. [35]	Porcine EVLP	Cytokine and gene profiles can characterize lungs on EVLP.
Iskender et al. [36,37]; Ghaidan et al. [38]	Porcine EVLP and Transplant	Removal of cytokines (e.g., IL-1a, IL-1b, IL-1ra, IL-4, IL-6, IL-8, IL-10, 1L-12, IL-18, TNF-a, INF-a, and INF-y) via adsorber improves organ quality.
Noda et al. [39] Stone et al. [40]	Rat and Porcine EVLP and Transplant Porcine	Removal of leukocytes can improve organ quality, diminish allorecognition, reduce T cell priming, and reduce T cell infiltration.
Noda et al. [41]	Rat EVLP and Transplant	Oxygenation of perfusate reduces inflammation activity in lungs.
Haam et al. [42,43]	Porcine EVLP and Transplant	EVLP gas composition affects resistance, pressures, and cytokines (IL-1b, IL-8, and TNF-a), and reduces inflammation (higher IL-10 and lower IL-6) of lungs.
Arni et al. [44]	Rat EVLP	EVLP lungs with sub-normothermic temperature with/without PFCOCs reduce inflammatory makers (TNFa, IL-6, and IL-7), chemokines controlling leukocyte movement (MIP-3a, MIP-1a, and GRO/KC), and growth factors (GM-CSF and G-CSF).
Nakajima et al. [45]	Porcine EVLP and Transplant	Lung lavage with surfactant replacement reduced inflammatory mediators (IL-1b, IL-6, and IL-8) and improved function (static pulmonary compliance, delta partial pressure of oxygen, and pulmonary vascular resistance) of organs on EVLP.
Lin et al. [7]	Porcine EVLP	A1AT administration to EVLP lungs reduced edema, apoptosis, inflammatory cytokines (IL-1a and IL-8) PAP, and PVR; improved compliance and gas exchange.
Yamada et al. [46]	Porcine EVLP and Transplant	NAC administration reduces inflammatory markers (IL-1b and MPO) for EVLP lungs.
Harada et al. [47]	Porcine EVLP and Transplant	Neutrophil elastase inhibitors administration reduces inflammatory markers (TNF-a, IL-6, and IL-8) and PVR; improves oxygen exchange and pulmonary compliance.
Allen [48], Hernandez-Lopez [49], Park [50], Liu [51]	Human Cell Model, Mouse Model	Engineered cells can tune proteins, sense specific compounds/tissues/or cells, migrate to geographically restricted regions, target and eliminate pathological tissue, and reduce fibrosis and improve cardiac systolic and diastolic function.
Martens et al. [52]	Porcine EVLP	Multipotent adult progenitor stem cells applied to lungs on EVLP reduces inflammatory markers (TNF-a, IL-1b, and INF-y).
La Francesca et al. [53]	Human EVLP	Multipotent adult progenitor cells applied to lungs on EVLP reduced immunological cells seen on BAL, inflammation, and edema.
McAuley [54]	Human EVLP	Multipotent mesenchymal stem cells applied to lungs on EVLP restored alveolar fluid clearance in injured lungs and improved endothelial permeability.
Stone et al. [55]	Mouse EVLP	Extracellular vesicles applied to lungs on EVLP produced an increase in IL-17, TNF-a, HMGB1, and CXCL1 inflammatory markers and inhibited neutrophil migration.
Nakakima et al. [56]	Porcine EVLP and Transplant	MSCs applied to EVLP lungs increased HGF production and increased IL-4 and decreased Caspase-3, IL-18, IFN-y, and the wet/dry weight of the lung.
Mordant et al. [57]	Porcine EVLP	MSCs applied to lungs on EVLP reduce IL-8.
Machuca et al. [58]	Porcine EVLP and Transplant	IL-10 gene therapy in donor lungs on EVLP improves gas exchange, reduces inflammation, and leads to lower TNF-a.
Hamid et al. [59]	Human EVLP	Aspirin reduces neutrophilic inflammation and alveolar injury in lungs on EVLP.
Martens et al. [60]	Porcine EVLP	Steroids administered to lungs on EVLP improved compliance, improved wet-to-dry weight, and reduced cytokines (IL-1b, IL-8, IFN-a, IL-10, TNF-a, and INF-y).
Charles et al. [61]	Porcine EVLP and Transplant	A2BAR administered to EVLP lungs improved compliance and reduced IL-12.
Cosgun et al. [62]	Porcine EVLP and Transplant	Trimetazidine administered to lungs on EVLP improved oxygenation/gas exchange, reduced myeloperoxidase, and reduced protein concentration in BAL.
Francioli et al. [63]	Rat EVLP	Pyrrolidine dithiocarbamate administered to EVLP lungs improved edema, reduced BAL protein concentration, and inhibited cytokines (TNF-a and IL-6).
Hsin et al. [64]	Human EVLP and Transplant	Metabolomics can be used to stratify lungs based on organ quality.

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
