# Peer review of "Transcriptomic Signatures in Lung Allografts and Their Therapeutic Implications"

_biomedicines, 2024, doi:10.3390/biomedicines12081793_

Round 1
Reviewer 1 Report
Comments and Suggestions for Authors
It is a very interesting paper, well-written. The authors are perhaps a little optimistic about point-of-care rapid diagnostic tools, but this demonstrates their confidence in the development of the technology. Congratulations
Author Response
Comment 1:
It is a very interesting paper, well-written. The authors are perhaps a little optimistic about point-of-care rapid diagnostic tools, but this demonstrates their confidence in the development of the technology. Congratulations
Response 1:
We thank the reviewer for the comments on the manuscript.
Reviewer 2 Report
Comments and Suggestions for Authors
The authors reviewed the current studies on transcriptomic analysis of lung transplants that had undergone ex vivo lung perfusion (EVLP). Below are a few comments that need attention:
1. The authors frequently emphasized the potential of coupling transcriptomic analysis and EVLP to improve the transplantation outcomes. However, they did not clarify if or how the potential has been realized in the studies reviewed.
2. It is recommended to include a summary graph that provides an overview of the molecular signatures identified in the studies mentioned in Table 1.
3. The second part of the article, titled “Leveraging EVLP as a platform to repair, recondition, and re-engineer lung allografts” does not align well with the title of the article. This section describes the strategies to improve EVLP, although the findings discussed in the text and table do not focus on transcriptomic results.
4. More discussion on perspective and future directions is suggested.
5. Please correct the formatting of the references in lines 398 and 400.
6. Please insert the figures and tables within the text instead of placing them at the end.
Author Response
The authors reviewed the current studies on transcriptomic analysis of lung transplants that had undergone ex vivo lung perfusion (EVLP). Below are a few comments that need attention:
- The authors frequently emphasized the potential of coupling transcriptomic analysis and EVLP to improve the transplantation outcomes. However, they did not clarify if or how the potential has been realized in the studies reviewed.
Response 1:
We thank the reviewer for this clarifying question. Indeed, the coupling of transcriptomic analysis and EVLP to improve transplantation outcomes has remains in the investigative phase and has not yet reached clinical practice. Transcriptomic analyses in lung transplant however have uncovered molecular pathways that could serve as targeted therapies for primary graft dysfunction (PGD). Our manuscript details this iterative process and highlights the potential for EVLP technology to implement such therapies. We have amended our manuscript to clarify this point in lines 240, 447, 463, and 471.
Comment 2:
- It is recommended to include a summary graph that provides an overview of the molecular signatures identified in the studies mentioned in Table 1.
Response 2:
We have added a figure 2 to summarize the findings outlined in Table 1.
Comment 3:
- The second part of the article, titled “Leveraging EVLP as a platform to repair, recondition, and re-engineer lung allografts” does not align well with the title of the article. This section describes the strategies to improve EVLP, although the findings discussed in the text and table do not focus on transcriptomic results.
Response 3:
As the reviewer mentioned, the field of EVLP as a targeted therapeutics platform is relatively new and not yet reached clinical practice, in particular, the on-demand use of transcriptional level data to implement targeted therapeutics. However, in this section we reviewed the pre-clinical studies evaluating therapies delivered on EVLP to date. As detailed in table 2 and in the text, these studies use as outcomes many of the molecular markers of inflammation that the previous section on transcriptomics highlighted. However, we do appreciate the reviewer’s noting that some of the included studies did not align with the molecular findings of the transcriptomics section. We have removed these references (previous 41, 43, and 50) from the amended manuscript in keeping with the overall objective of the review paper.
Comment 4:
- More discussion on perspective and future directions is suggested.
Response 4:
We have amended the conclusion to enhance the discussion of current state of affairs regarding EVLP in clinical use. We highlight the future potential impact to further elucidate the molecular pathways underlying lung allograft dysfunction, predict graft function, and become a platform for targeted interventions. We also discuss eventual addition of metabolomic analysis to further enhance the potential of this technology to improve outcomes. Finally, we discuss the holy grail of on-demand therapeutics based on the multi omics analysis which will require the future development of rapid diagnostics. (lines 447, 463, and 471)
Comment 5:
- Please correct the formatting of the references in lines 398 and 400.
Response 5:
Thank you, we have fixed this formatting.
Comment 6:
- Please insert the figures and tables within the text instead of placing them at the end.
Response 6:
Thank you, we have placed the figures in the text.